# Recent Advances in the Photoreactions Triggered by Porphyrin-Based Triplet–Triplet Annihilation Upconversion Systems: Molecular Innovations and Nanoarchitectonics [note 1]

**DOI:** 10.3390/ijms23148041

**Published:** 2022-07-21

**Authors:** Bin Yao, Hongfei Sun, Youzhou He, Song Wang, Xingyan Liu

**Affiliations:** Chongqing Key Laboratory of Catalysis and New Environmental Materials, College of Environment and Resources, Chongqing Technology and Business University, Chongqing 400067, China; yaobinpku@126.com (B.Y.); sunhongfei@ctbu.edu.cn (H.S.); yzhectbu@163.com (Y.H.); wangsong@ctbu.edu.cn (S.W.)

**Keywords:** porphyrins, triplet–triplet annihilation upconversion, photoisomerization, photocatalysis, photopolymerization, photodegradation, water splitting, biomedical applications

## Abstract

Triplet–triplet annihilation upconversion (TTA-UC) is a very promising technology that could be used to convert low-energy photons to high-energy ones and has been proven to be of great value in various areas. Porphyrins have the characteristics of high molar absorbance, can form a complex with different metal ions and a high proportion of triplet states as well as tunable structures, and thus they are important sensitizers for TTA-UC. Porphyrin-based TTA-UC plays a pivotal role in the TTA-UC systems and has been widely used in many fields such as solar cells, sensing and circularly polarized luminescence. In recent years, applications of porphyrin-based TTA-UC systems for photoinduced reactions have emerged, but have been paid little attention. As a consequence, this review paid close attention to the recent advances in the photoreactions triggered by porphyrin-based TTA-UC systems. First of all, the photochemistry of porphyrin-based TTA-UC for chemical transformations, such as photoisomerization, photocatalytic synthesis, photopolymerization, photodegradation and photochemical/photoelectrochemical water splitting, was discussed in detail, which revealed the different mechanisms of TTA-UC and methods with which to carry out reasonable molecular innovations and nanoarchitectonics to solve the existing problems in practical application. Subsequently, photoreactions driven by porphyrin-based TTA-UC for biomedical applications were demonstrated. Finally, the future developments of porphyrin-based TTA-UC systems for photoreactions were briefly discussed.

## 1. Introduction

Photon upconversion, as a promising anti-stokes shift technology that could convert photons from low-energy to high-energy ones, has attracted growing attention from researchers owing to its wide applications in various fields such as photovoltaics, photocatalysis, sensing, nonlinear optics, and photodynamic therapy [1,2,3,4,5,6,7]. In the past few decades, lanthanide-doped upconversion nanoparticles (UCNPs) [8,9,10,11,12,13,14,15,16,17,18,19] and inorganic–organic hybrid nanomaterials [20,21,22,23] have been widely investigated as upconversion materials, and several upconversion mechanisms (such as excited state absorption, energy transfer upconversion and cooperative sensitization upconversion) have also been addressed [12]. However, these inorganic upconversion materials possess certain shortcomings such as a relatively high excitation power as well as low upconversion quantum yield. Moreover, the toxicity of lanthanide ions in living organisms also needs to be further explored. Developing organic upconversion materials is an urgent task.

As a novel alternative upconversion pattern, triplet–triplet annihilation upconversion (TTA-UC) is primarily based on organic upconversion materials [24,25,26,27,28,29,30]. The TTA-UC tends to be a bimolecular system composed of an organometallic complex and a conjugated hydrocarbon, which act as a sensitizer (donor) and an annihilator (emitter/acceptor), respectively. The specific mechanism of TTA-UC is provided in Figure 1 [31]. Initially, the sensitizer absorbs a low-energy photon and then becomes excited to its singlet excited state. The metal-to-ligand charge-transfer (MLCT) feature of the sensitizer leads the singlet excited state to undergo intersystem crossing (ISC) to reach its long-lived and more stable triplet excited state. Under suitable energy levels and distance, the sensitizer and annihilator undergo triplet–triplet energy transfer (TTET), resulting in the triplet excited state of the annihilator (^3^An*). Two closer ^3^An* undergo triplet–triplet annihilation, leading to one annihilator becoming excited to a high-energy singlet excited state (^1^An*) and the other losing its energy and returning to the ground state. ^1^An* can also return to its ground state by releasing upconversion fluorescence. Compared with lanthanide-doped UCNPs, TTA-UC possesses the advantages of a low excitation power density, large absorption efficiency, tunable excitation as well as emission wavelength, and high quantum yield [32].

Both the sensitizer and annihilator are important for the TTA-UC process. The sensitizer, which is the starting material and is involved in most of the process, has been paid more attention. Various organometallic complexes including Ru complexes [33,34,35], Ir complexes [36,37,38], Os complexes [39,40,41] and porphyrin derivatives [42,43,44,45] were extensively investigated as sensitizers. Among these organometallic complexes, porphyrin derivatives are most often used in comparison to other counterparts. The planar conjugated configurations provide porphyrins with favorable optical and electrical characteristics, such as an intense absorption of visible light, and facilitate electron transfer [46]. Porphyrin rings could be complexed with various metal ions bearing different valence states. On one hand, the lifetime and proportion of the triplet state can be well regulated; on the other hand, porphyrin rings are helpful in stabilizing various intermediates in the upconversion process. Additionally, porphyrins are involved in various biological processes, and they should have good biocompatibility [47]. Moreover, the peripheral positions of porphyrin rings are readily derivable, including both *meso*- and *β*-positions [48]. In particular, the regulation of the conjugation length at the *β*-positions drastically changes the absorption wavelengths of porphyrins, such as Pt/PdOEP, Pt/PdTPTBP, and Pt/PdTPTNP for green light, red light, and near-infrared (NIR) light, respectively (Figure 2). By combining them with annihilators (i.e., DPA and perylene, Figure 2 lists the commonly used annihilators) possessing different emission wavelengths, the porphyrin-based TTA-UC systems can achieve the upconversion of various wavelength ranges including green-to-blue, red-to-blue, NIR-to-blue, and so on, making metalloporphyrins excellent sensitizer candidates for various applications [49].

As a new upconversion technology, TTA-UC developed rapidly in recent years. Many related issues have been widely investigated, such as how to improve upconversion efficiency [50,51], how to overcome oxygen quenching [52,53], and how to achieve solid-state applications [54,55,56,57,58]. Furthermore, a great many TTA-UC-based examples for practical applications have also emerged, including solar cells [59,60], bioimaging [61,62], circularly polarized luminescence [63,64,65], sensing [66,67], and photoreactions [31]. Especially for photoreactions triggered by porphyrin-based TTA-UC systems, great achievements of molecular innovations and nanoarchitectonics [68] have been witnessed in the past few years. For example, NIR-to-blue upconversion is demonstrated to be able to drive chemical transformations [69], and TTA-UC-assisted photopolymerization has been successfully applied for 3D printing [70]. Surprisingly, although many papers reviewed TTA-UC materials and their applications [27,28,31], no specific reviews focused on the photoreactions triggered by porphyrin-based TTA-UC systems. In consideration of the significant contributions of porphyrin-based systems to the photochemical applications of TTA-UC as well as to inspire more scientists to devote themselves to this meaningful field, this review systematically summarized the recent developments in the photoreactions facilitated by porphyrin-based TTA-UC systems, and the molecular innovations and corresponding nanoarchitectonics are emphatically discussed.

## 2. Photochemistry of Porphyrin-Based TTA-UC for Chemical Transformations

Let us refer back to Figure 1 regarding the TTA–UC process; in addition to emitting a high-energy photon through upconverted fluorescence, there are two ways in which the ^1^An* can be involved in a photochemical reaction [31]. On one hand, it can act as an intermediary to transfer its energy to another photocatalyst to initiate a photochemical reaction. On the other hand, the ^1^AN* could also serve as a photocatalyst for single electron transfer (SET) to directly catalyze a photoreaction [71,72,73]. Two alternative methods provide great convenience, and various types of photochemical reactions have been successfully achieved using porphyrin-based TTA-UC technology. In the following sections, according to different reaction types, the applications of porphyrin-based TTA-UC systems for chemical transformations are discussed in detail.

### 2.1. Photoisomerization

Photoisomerization plays an important role in many stimulus-responsive materials, and many functional groups possess this property, such as N=N, C=C, and C=N double bonds [5,66,74,75]. Crosslinked liquid-crystal polymers (CLCPs) are well-known light-driven soft actuators containing these motifs. Nevertheless, the modulating wavelengths are usually short-wavelength ultraviolet or blue light, so they face the issues of safety and cost in practical applications. In 2013, Yu, Li et al. reported a solution using TTA upconverted fluorescence as the modulating light [76]. As shown in Figure 3A, the soft actuator was composed of two layers combined with transparent adhesive. The upper layer was the upconversion film that mainly contained a rubbery polyurethane polymer doped with PtTPTBP/BDPPA TTA-UC couple, and the lower layer was an azotolane-containing CLCP film. Upon excitation at 635 nm light, the upconversion film underwent red-to-blue TTA-UC, and the blue upconverted fluorescence was utilized to drive the *trans*−*cis* isomerization of azotolane groups after an emission−reabsorption process. Concomitantly with photoisomerization, the orientation of mesogens adjacent to the upconversion film were rearranged, leading to the macroscopic bending of the soft actuator toward the light source. When turning off the light, the bending was retained sufficiently until thermal manipulation returned the system to its original state. Compared with the photoisomerization directly induced by blue light, the porphyrin-based TTA-UC system possessed the merits of low power density and long wavelength. Notably, this work is the first to demonstrate that TTA-UC can be utilized to perform macroscopic mechanical work.

For conventional TTA-UC systems, the electron transfers between annihilators and substrates are usually intermolecular, in which the upconversion light undergoes the emission−reabsorption process, leading to inefficient reactions. In 2019, Abe and coworkers innovatively assembled an annihilator and substrate into one molecule via a covalent bond, which successfully achieved a high-efficiency photochromic reaction owing to the distinguishable conjugation lengths of different isomers (Figure 3B) [77]. In their study, a perylene moiety was covalently bonded to the 2-position of the imidazole ring of RPIC, leading to the resulting Pery-RPIC which could induce photoisomerization upon excitation by ultraviolet light [78]. By means of the concept of TTA-UC with PdTPTBP as the sensitizer, the photoisomerization could proceed by excitation with red light. With the aid of femtosecond time-resolved spectrum techniques to monitor the intermediate processes, the SET from the perylene subunit to the photochromic subunit was clearly demonstrated. Impressively, this report is the first example of a photochromic reaction which directly utilized the high energy of ^1^AN*, providing a good reference for the high-efficiency utilization of the energy of TTA-UC via the covalent bonding mode.

### 2.2. Photochemical Synthesis

Using light to drive photochemical synthesis is a very attractive method owing to its directional, cheap, clean, and tunable characteristics [79,80,81,82,83]. Regarding the organic chemical reactions triggered by TTA-UC systems, the biggest challenge might be molecular oxygen because it quickly quenches the triplet species. In 2012, Kim et al. developed a microcapsule strategy to circumvent the problem, in which the sensitizer (PtOEP) and annihilator (DPA) dissolved in a hexadecane/polyisobutylene (HD/PIB) mixture were encapsulated inside rigid, spherical, and monodispersed polymer microcapsules [84]. The HD/PIB mixture could effectively prevent oxygen quenching, so the TTA-UC process could be performed in ambient aqueous environments. Subsequently, the TTA-UC microcapsules were dispersed in an aqueous phase system that contained Pt/WO_3_ for photocatalytic experiments. Upon irradiation by green light (*λ* = 532 nm), the system initiated a semiconductor photocatalytic feature that generated •OH to oxidize coumarin to 7-hydroxycoumarin (entry 1 in Table 1) by means of a TTA-UC process. Gratifyingly, the photon energy of excitation light was lower than the bandgaps of any existing semiconductor photocatalysts at that time. Based on this finding, the excitation wavelength was further extended to red light and the particle size was reduced from microcapsule to nanocapsule by the same group [85]. A larger conjugated sensitizer (PdTPTBP) and annihilator (perylene) were nanoscale encapsulated within a rigid silica shell, the surface of which was further functionalized with CdS nanoparticles. The system showed stable red-to-blue upconversion under oxygen-rich aqueous conditions and was also demonstrated to be the catalyst for sub-bandgap photocatalytic oxidation of coumarin (entry 2 in Table 1). A series of electron and energy transfer processes together contributed to the photocatalytic oxidation. Except for the TTA-UC process in the silica shell, the •OH derived from superoxide radical anion intermediate (O_2_^•−^), which originated from the transference of one electron from the conduction band of CdS to molecular oxygen, was also supposed thought to play a critical role in the photocatalytic process. 

In 2015, Díaz et al. reported another alternative approach that could solve the highly sensitive problem of molecular oxygen [86]. In their study, PtOEP and DPA were selected as the sensitizer and emitter, respectively, and together with aryl bromides they were embedded in supramolecular gels to test for the photoreduction of aryl halide (entry 3 in Table 1), which was generally believed to be inactive under visible light [88]. The organogel system exhibited green-to-blue upconversion upon excitation by green light and showed enhanced conversion efficiency, good mass balance as well we high overall yields under aerobic and room temperature conditions in comparison to the dissolved PtOEP and DPA. Besides, the gelator was also proved to not participate in the photoreaction, but played a key role as a nanoreactor. The possible mechanism for the photoreduction is outlined in Figure 4. Upon 532 nm irradiation, PtOEP and DPA underwent a TTA-UC process to produce an excited singlet of DPA (^1^(DPA)*), and the following process of SET to aryl halide resulted in an unstable radical anion ArX^•−^. The ArX^•−^species occurred in cleavage reaction to afford Ar^•^, which underwent the final H-atom transfer (HAT) to yield the reduction product Ar-H. This work confirms that low-weight molecular gelators could be utilized as critical confined media for photoreactions, providing important supports for more demanding photophysical processes.

In view of the intrinsic challenges of the utilization of visible-light irradiation, such as its short penetrating length as well as competition between reactants and photocatalysts, an extension of the operating wavelength of TTA-UC to the NIR region is considered highly necessary. NIR-induced reactions via lanthanide-doped UCNPs have been reported on extensively [89]. Nevertheless, it was not until 2019 that Congreve, Rovis and Campos et al. successfully achieved this great breakthrough using a porphyrin-based TTA-UC system [69]. A metalloporphyrin with a larger conjugate length, PtTPTNP, was selected as a sensitizer while tetratertbutylperylene (TTBP) acted as an annihilator. Upon irradiation by NIR light (*λ* = 730 nm) in anaerobic solutions, the sensitizer/annihilator exhibited NIR-to-blue photon upconversion with a large anti-Stokes shift (1.0 eV) and moderate upconversion yield (2%). The larger anti-Stokes shift rendered the TTA-UC system suitable for driving various reactions. On one hand, the TTA-UC could be applied to catalyze small-molecule organic reactions, such as transferring the high energy of ^1^TTBP* to Ru(bpy)_3_(PF_6_)_2_ for a [2 + 2] cyclization reaction or directly as a photocatalyst for vinyl azide sensitization reactions (entry 4 in Table 1). On the other hand, the NIR-to-blue photon upconversion could also be utilized to initiate radical polymerization including both homopolymerization and copolymerization (entry 1 in Table 2, the application in photoinitiated polymerization will be expounded on in detail in the following section). 

Han’s group paid close attention to the low efficiency of NIR-light-activated TTA-UC [87]. Generally, the TTA-UC process contains several triplet excited states (^3^Sens* and ^3^An*) and singlet excited states (^1^Sens* and ^1^An*), and their energy levels have decisive effects on the upconversion efficiencies. Different from other studies that mainly focused on the sensitizer optimizations, they adopted the strategy of optimizing the annihilators. As depicted in Figure 5, a series of perylene derivatives were designed and synthesized, including an alkyl annihilator(Py0), three monoaryl-alkynyl annihilators (Py1−Py3), and two diaryl-alkynyl annihilators (Py4−Py5). From Py0 to Py5, the conjugation length of the molecular skeleton increased gradually, leading to decreased ^3^An* and ^1^An* energy levels. Especially for Py4 and Py5, the ^3^An* energy levels were lower than that of the singlet excited state of sensitizer (^1^PdTPTNP*), better facilitating the TTET process since the unfavorable endothermic process had changed to a more favorable exothermic one. Therefore, the TTA-UC efficiencies were greatly improved (for Py4 and Py5 TTA-UC systems, 7.6% and 16.7% at 653 nm light excitation, and 6.4% as well as 14.1% at 720 nm light excitation). Furthermore, the PdTPTNP/perylene derivative pairs were combined with a visible light absorbing photocatalyst, and the NIR-driven photooxidation of aryl boronic acid to aryl phenol was investigated in anaerobic solutions using these composite catalysts. The PdTPTNP/Py5 couple bearing the biggest TTA-UC efficiency was observed to behave with the best photocatalytic performance, implying that the increase in the TTA-UC efficiency could enhance the photocatalytic reaction yield, and indicating the success of the annihilator optimization strategy. 

Great achievements were made in terms of photochemical reactions driven by NIR-based TTA-UC systems; however, these studies also presented some drawbacks such as the experiments needed to be carried out under anaerobic conditions. More efforts are needed to improve the tolerance of the system to molecular oxygen, and the abovementioned low weight molecular supramolecular gel as well as micro/nanocapsule strategies may provide some references.

### 2.3. TTA Photopolymerization

As mentioned above, upon excitation by NIR light, the NIR-to-blue TTA-UC of the PtTPTNP/TTBP couple could be applied not only to drive photochemical synthesis but also to initiate radical polymerization [69]. Indeed, this is not the first case of radical polymerization initiated by NIR light upconversion [90,91,92,93], but it is the first report based on TTA-UC which shows the advantages of high quantum efficiency and a low excitation threshold. The operation of TTA photopolymerization (TTAP) was very simple, and directly irradiated the sensitizer, annihilator as well as initiator in a neat monomer using NIR light. The photoinitiated mechanism can be explained as follows: the singlet excited state of TTBP (^1^TTBP*) possessed a strong reducing nature, which could be used to initiate radical polymerization by the C–Br bond reduction of the initiator. In order to demonstrate the penetration ability of NIR radiation, TTAP was conducted behind various barriers that could absorb visible light or on a multi-gram scale. Both of the above operations could achieve good polymerization results with NIR irradiation, but this was not the case for directly using blue light. In a recent report, Fang et al. also demonstrated TTA-UC-assisted photopolymerization for different monomers using visible light (entry 2 in Table 2), in which the photocuring polyurethane acrylates (PUA) could be utilized as excellent adhesion materials for plastic substrates [94].

**Table 2 ijms-23-08041-t002:** Summarized photoinitiated polymerizations via porphyrin-based TTA-UC.

Entry	Sensitizer	Annihilator	Exemplary Reactions	Ref.
1	PtTPTNP	TTBP	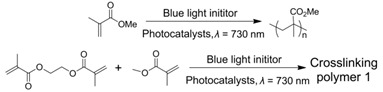	[69]
2	PtOEP	DPA	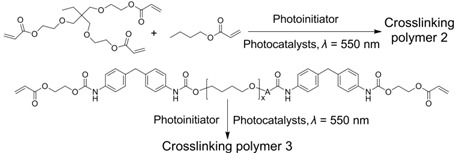	[94]
3	ZnTPP	ZnTPP	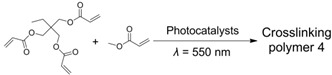	[95]
4	PdTPTBP	TIPS derivatives	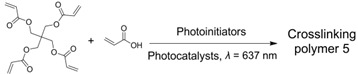	[70]
5	PdOEP	DPA	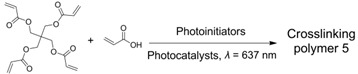	[96]

Interestingly, metalloporphyrins such as ZnTPP were leveraged for visible light photoinitiated polymerization such as reversible addition−fragmentation chain transfer (RAFT) living radical polymerization [97,98,99,100]. However, extra photoinitiators (such as RAFT agents) were required, and the photoinitiated mechanism was based on the photoinduced electron transfer (PET) from ^3^ZnTPP* to RAFT agents, leading to the production of free radical species to initiate polymerization [101]. In 2020, Castellano et al. proved that the homomolecular TTA-UC [102,103] could also be leveraged to initiate free-radical polymerization using green light (550 nm) without any co-initiators (entry 3 in Table 2) [95]. The mechanism of the homomolecular TTAP is depicted in Figure 6. Regarding ZnTPP, after singlet excitation and the ISC process, the molecule was in its triplet excited state (^3^ZnTPP*). Compared with conventional TTA-UC, the TTET process was nonexistent in this system, and two ^3^ZnTPP* species occurred in TTA to provide an S_2_ excited state of the ZnTPP molecule (^1^ZnTPP*’). Afterwards, ^1^ZnTPP*’ and monomers, such as methyl acrylate (MA) or trimethylolpropane triacrylate (TMPTA), underwent dynamic electron transfer to provide a monomer free radical to initiate polymerization. Considering the simplicity of the whole polymerization system, which only requires ZnTPP, monomers, and green light, the controllability of polymerization can be greatly improved, providing convenience for the preparation of various soft materials. 

Given the great advantages of TTAP, Congreve et al. innovatively applied it in the 3D printing field (the photopolymerization shown as entry 4 in Table 2) [70], which was meaningful to circumvent the problems of conventional or two-photon absorption photopolymerization in 3D printing, such as a higher excitation power, lower penetrating ability and lower tenability [104,105,106]. As represented schematically in Figure 7A, in order to avoid the leakage of components as well as the aggregation-induced light scattering of particles, the upconversion materials were assembled into the cores of silica nanocapsules bearing a core–shell configuration with PEG segments grafted around the periphery. With PdTPTBP serving as the sensitizer, a series of annihilators (Figure 7B) substituted with different halogen atoms at the anthracene were selected to systematically modulate the threshold light intensity of upconversion materials. From TIPS-A to 2Br-TIPS-A, the threshold values varied from 1.7 W·cm^−2^ to 283 W·cm^−2^, which could be applied for different printing modes such as in a parallel excitation printer (for the former three annihilators) and monovoxel excitation printer (for Br-substituted ones). The red-to-blue upconversion feature in the UCNPs was demonstrated using spectroscopy techniques. After incorporating the UCNPs into 3D printing resins containing commercially available monomers and photoinitiators, they attempted to print products in different shapes using different printing modes (Figure 7C shows the monovoxel printing mode for a benchmark boat), all of which achieved excellent 3D printing performances. Remarkably, the excitation power densities of these TTA-UC-assisted 3D printings were several orders of magnitude lower than those of two-photon-based 3D printings. Notably, when this article was under review, the Hayward group also found that porphyrin-based TTAP (entry 5 in Table 2) could be effectively utilized for high-resolution 3D printing (~100 nm) [96]. Doubtlessly, these studies not only present new applications for TTA-UC but also provide a novel approach for high-resolution 3D printing.

### 2.4. Photocatalytic Degradation

Semiconductor photocatalysts (such as TiO_2_, WO_3_, Bi_2_WO_6_ and CdS) can absorb sunlight and generate electrons and holes to provide reactive oxygen species (ROS) to decompose pollutants in air or water solutions, which is meaningful to solve the increasingly serious environmental pollution problems [107,108,109,110]. Nonetheless, most existing semiconductor photocatalysts are only responsive to light at short wavelengths (<480 nm), which occupies only a tiny fraction of solar energy. In order to realize the efficient utilization of solar energy and to improve photocatalytic efficiency, combining upconversion with semiconductor photocatalysts provides a good approach [111,112].

Kim et al. firstly reported the photocatalytic degradation of acetaldehyde for air purification by utilizing low-energy photons (532 nm) with the help of porphyrin-based TTA-UC system [113]. As revealed in Figure 8A, a device with a double-layer structure was assembled in their study. The upper layer was the TTA-UC layer, the active components of which were mainly PtOEP and DPA dispersed into inert rubbery polyurethane polymer. Besides, optimized 20% AgNP-SiO_2_ particles were also incorporated into the film since they could enhance the absorption of PtOEP via the local surface plasmon resonance (LSPR) mechanism [114,115,116]. The lower layer was the photocatalyst layer containing nanodiamond (ND) loaded with WO_3_. Upon excitation by green light, the ND/WO_3_ composite photocatalyst was able to absorb the upconverted fluorescence generated from the TTA layer, thus catalyzing the degradation of acetaldehyde to CO_2_. Successfully, 20 ppmv acetaldehyde was completely degraded within 3 h using a double-layer device. Following this work, the same group further designed a dual-layer TTA-UC device to improve the energy utilization efficiency (Figure 8B), which could upconvert both low-energy red (PdTPTBP/perylene couple) and green light (PtOEP/perylene couple) to high-energy blue light [117]. Upon white LED irradiation, the photocatalytic degradation efficiency of the device with dual-layer TTA-UC films was twice that of its single-layer counterparts.

Inspired by the plasmon-enhanced TTA-UC approach, Lu, Wang et al. designed a multilayer composite photocatalyst for tetracycline degradation [118]. The detailed composition and catalytic mechanism are described in Figure 8C. For the optimized 0.3 AuNPs-PtDPAP/G/CdS photocatalyst (the composition of each layer see the Abbreviations section), the role of each layer was plasmon-enhanced with a TTA-UC layer, adhesive layer, and semiconductor photocatalyst layer, respectively. Besides, the excellent electron transfer feature of graphene could also depress the recombination of photogenerated electrons and holes in the CdS layer. Upon excitation by green light or visible light, the upconverted high-energy photons from the 0.3 AuNPs-PtDPAP could effectively sensitize the CdS catalyst for photocatalytic tetracycline degradation. The degradation process was confirmed to comply with the law of pseudo-first-order reactions (rate constant *k*_pfo_ = 0.294 h^−1^). Furthermore, the composite catalyst showed good recyclability with no significant decrease in catalytic performance after five recycles, and thus the secondary contamination of catalysts was largely avoided. On the basis of these results, photonic crystals (V-shape anodic Al_2_O_3_ templates were used in this study) were further incorporated to collectively enhance photocatalytic performance by the same group (Figure 8D) [119]. Upon light irradiation, the inverted V-shape (AVS) structure of the photonic crystal could stimulate a Bloch surface wave and was then coupled with excitons in a quantum well to improve the surface field; therefore, they could be utilized to enhance upconverted luminescence via photon localization [120,121]. Through the synergistic effect of photonic crystal and AuNPs plasmon resonance, the upconverted luminescence of the AVS-0.3 AuNP-PtDPAP film was enhanced 7.68 times in comparison to pristine PtDPAP. The upconverted system was evaluated for photocatalytic tetracycline degradation after assembly with *g*-C_3_N_4_@CdS film. Compared with the pristine PtDPAP system, the AVS-0.3 AuNP-PtDPAP gained a 3.88-fold enhancement in *k*_pfo_, implying the success of the hybrid effect of photonic crystal and plasmon resonance.

The abovementioned examples of photocatalytic degradation are all of heterogeneous photocatalysis. Li’s group recently demonstrated the homogeneous photocatalytic feature of porphyrin-based TTA-UC for the degradation of an oxidized lignin model (Figure 9A) [122]. Two sensitizer/annihilator pairs were tested (Figure 9B), in which PdTPNEt2P possessed lower oxidative potential and greater conjugation length compared with PdTPP. With *N*,*N*-Diisopropylethylamine (DIPEA) serving as the sacrificial reagent, upon excitation by green light in DMF solutions, the degradation efficiency of PdTPNEt2P/perylene TTA-UC pair (98.94%) was more than 13 times that of PdTPP/perylene (7.21%). The enhanced degradation efficiency might be ascribed to the longer triplet lifetime of PdTPNEt2P, which could facilitate the diffusion-controlled TTET process. In addition to the near quantitative degradation efficiency, the PdTPNEt2P/perylene pair also exhibited high degradation selectivity with an 87.81% yield of guaiacol (51.16% of pristine perylene catalyst for comparison), implying that low-energy light was beneficial to avoid side reactions. For two different TTA-UC pairs, the possible photophysical processes for the generation of active species in the degradation processes are depicted in Figure 9C,D, respectively. In contrast to the highly quenched TTET process of DIPEA in the PdTPP/perylene pair, the lower oxidative potential of PdTPNEt2P made the TTET process more effective, and the subsequent PET from DIPEA to the singlet state of perylene provided a perylene radical anion (pe^•−^) to act as the real catalytic active species. Considering that the absorption, emission, and energy levels could be finely regulated by selecting different sensitizer/annihilator pairs, the porphyrin-based TTA-UC may show promising prospects in lignin photocatalytic decomposition with high efficiency and high selectivity, which is of significance for the extraction of higher value products from renewable resources.

### 2.5. Photochemical/Photoelectrochemical Water Splitting

As discussed above, the main feature of TTA-UC is to transform low-energy light into high-energy upconverted fluorescence. Thus, the solar energy utilization efficiency, especially for a long-wavelength range, could be greatly enhanced. Thanks to this unique property, in addition to environmental remediation, TTA-UC also plays a pivotal role in energy conversion and storage [123,124,125,126]. The photocatalytic splitting of water into H_2_ and O_2_ is a promising technique that can convert solar energy to chemical energy. Successfully, the TTA-UC technique has been introduced in photocatalytic water splitting systems to improve energy utilization efficiency as well as photocatalytic performance.

Lu et al. firstly combined the concept of TTA-UC and photocatalytic hydrogen evolution reaction (HER) [127]. The compositions and photocatalytic mechanism of the PtDPAP@SiO_2_@CdS/Pt photocatalyst are revealed in Figure 10A. PtDPAP was dispersed in oil acid to guarantee effective diffusion and then was encapsulated into rigid silica shells to isolate molecular oxygen. After that, the CdS photocatalyst was deposited on the silica shell followed by the deposition of the Pt co-catalyst to facilitate the separation of photoinduced electron–hole pairs. With triethanolamine (TEA) serving as the sacrificial reagent and excitation by visible light or green light, the composite photocatalyst was constructed as a heterogeneous catalysis for HER in aqueous solution. Despite the low rate of hydrogen production (22.7966 μmol/h upon visible light) and low quantum efficiency (0.0777% upon green light), this study confirmed the feasibility of the application of TTA-UC for photocatalytic water splitting. 

In order to improve the photocatalytic HER rate and quantum efficiency, Lu’s group further optimized their catalytic system (Figure 10B) [128]. Au nanoparticles and PtDPAP were collectively encapsulated into the silica shell to elevate upconversion efficiency via a plasma resonance mechanism. In addition, graphitic carbon nitride (*g*-C_3_N_4_) was also introduced to form heterojunctions with CdS, resulting in good dispersibility and enhanced chemical stability. Moreover, the suitable energy levels between *g*-C_3_N_4_ and CdS facilitated the separation of electron–hole pairs as well as electron transfers. After surface functionalization, Au-PtDPAP@SiO_2_ was immobilized on *g*-C_3_N_4_-CdS nanosheets via electrostatic interactions. When the composites were conducted as photocatalysts for HER experiments, the optimized catalyst Au-PtDPA@SiO_2_@0.5NC2 revealed high hydrogen production (ΔH_2_ = 2.76 mmol·g^−1^·h^−1^) upon visible-light, high apparent quantum efficiency (1.439%) upon green light, and relatively good stability (>90% after 20 runs).

Aside from photocatalytic HER, TTA-UC has also been tested for the enhancement of the photocatalytic performance of a more complicated and challenging oxygen evolution reaction (OER) [129], since it requires a theoretically high potential (1.23 V) and contains four electron transfer processes [130]. Given that triplet species are easily quenched by molecular oxygen, as depicted in Figure 11, the sensitizer PtOEP and emitter DPA together with a cetyltrimethylammonium bromide surfactant were intercalated in a layered clay compound (saponite). On one hand, the layered structure of saponite and the co-intercalated surfactant effectively blocked molecular oxygen. On the other hand, their transparency in the visible light region led them to have little effect on the TTA-UC process. When the upconverting composite and a short-wavelength-light responsive Mo-doped BiVO_4_ (Mo:BVO_4_) photocatalysts were dispersed in the same aqueous solution with AgNO_3_ as the sacrificial electron acceptor and excitation by green light, the upconverted blue fluorescence successfully sensitized the Mo:BVO_4_ photocatalyst to induce water splitting. The photocatalytic OER rate was enhanced from 0.38 mmol·h^−1^ to 0.49 mmol·h^−1^ in the presence of the upconverter, demonstrating that the highly sensitive TTA-UC process could even be applied for photocatalytic OER and the success of the intercalated strategy.

Evidently, both photocatalytic HER and OER have inevitable shortcomings such as the need for additional sacrificial reagents, which limits their practical applications. Photoelectrochemical water splitting circumvents this problem, in which an external bias potential is applied in combination with illumination [131,132,133]. In fact, in 2012, Castellano’s group conducted proof-of-concept experiments that demonstrated the upconversion-driven photoelectrochemical cell based on a PdOEP/DPA TTA-UC pair [134]. However, the sensitizer and annihilator were sealed in a cuvette containing degassed toluene, which did not make much sense for practical applications. In 2017, Meinardi et al. designed a novel photoelectrochemical water splitting cell enhanced by TTA-UC [135]. As represented in Figure 12A,B, with WO_3_ as the photocatalyst, PtOEP and DPA doped in poly(octyl acrylate) were immobilized on the back of the WO_3_ layer to harvest the transmitted photons. Additionally, a layer of poly(lauryl methacrylate) doped with CdSe/ZnS nanocrystals was introduced to the back of the TTA-UC layer to act as a boosting layer (BST) and an aluminum layer was finally applied to finish the photoanode as a black mirror to diminish photon loss. Their absorption spectra and fluorescence emission spectra were in good agreement (Figure 12C), and thus the incident photons were utilized by the photocatalyst. Upon testing for photoelectrochemical water splitting with sunlight irradiation and a bias voltage of +0.9 V, the cell containing the TTA-UC layer demonstrated better performance. When irradiated by green light, the upconverted photons were found to contribute to more than 80% of the photocurrent, verifying the ability of TTA-UC to trigger photoelectrochemical reactions. The photocathode had four different layers in this study, which indicates that it was too complicated a process. In 2019, Moon et al. simplified the photocathode structure, which contained only two layers (Figure 12D) [136]. With Mo:BiVO_4_ serving as photocatalyst, a luminescent back reflector (LBR), which was composed of PdTPTBP, perylene, TiO_2_ nanoparticles as well as PU polymer, was immobilized on the back of a photocatalyst (Figure 12D). The introduction of TiO_2_ nanoparticles was supposed to reduce total internal reflection and enhance surface emission. Under the same conditions, the photocurrent of BiVO_4_/LBR photoanode gained a 17% enhancement in contrast to that of the bare BiVO_4_ photoanode (Figure 12E).

## 3. Photochemistry of Porphyrin-Based TTA-UC for Biomedical Applications

The merits of porphyrin-based TTA-UC systems, including the intense absorption of porphyrin sensitizers, low excitation power, high upconversion quantum yield and tunable excitation/emission wavelengths, make them useful not only in chemical transformations but also in biomedical applications. They have been widely investigated as key components in bioimaging [137,138,139,140,141] and biosensing [142,143,144]. Except for these applications, the use of porphyrin-based TTA-UC for triggering photoreactions in biological systems are also emerging, playing an active role in drug release.

Photoactive Ru complexes are classical carriers for prodrugs in photoresponsive anticancer therapy; however, the characteristic of blue light stimulus-responsiveness places their operating wavelength (400–500 nm) outside the “phototherapeutic window” (600–1000 nm). In 2014, Bonnet’s group extended the operating wavelength to the red light region with the aid of the TTA-UC technique [145]. Ru^2+^-SRR’ containing a thioether-cholesterol ligand was selected as the target Ru complex (structures shown in Figure 13A). Together with the PdTPTBP/perylene TTA-UC pair, they were collectively intercalated into a PEGylated liposome which was similar to the environment of the phospholipid bilayer of the cell membrane (Figure 13B). Under deoxygenated conditions and irradiation using a photodynamic therapy (PDT) laser at 630 nm, the red-to-blue upconverted fluorescence successfully triggered the hydrolysis of Ru^2+^-SRR’ to release Ru^2+^-OH_2_; therefore, the remaining SRR’ ligands were able to smoothly diffuse from the membrane. Following this, the detailed mechanisms of the TTA-UC-assisted photodissociation were systematically investigated [146]. As outlined in Figure 13C, TTA-UC firstly generated the singlet state of perylene, which underwent intermolecular Förster resonance energy transfer (FRET) to a singlet state of Ru^2+^-SR_2_. After the production of the ISC and thermal population, a triplet MLCT state as well as a triplet metal-centered (MC) state were obtained, respectively. Eventually, the SRR’ ligand was replaced by a H_2_O molecule to complete the photosubstitution reaction. Herein, two critical processes, TTA-UC and FRET, were strategically combined to activate the photodissociative ruthenium complex in this system. 

The metal–ligand photodissociation regulated by upconverted fluorescence represents a potentially promising application of TTA-UC-assisted photoactivatable chemotherapy in oxygen-poor diseased tissues such as hypoxic tumors. Undoubtedly, the highly sensitive oxygen quenching problem of TTA-UC needs to be addressed. In 2015, Kohane’s group developed PLA−PEG micellar nanoparticles to solve this problem and achieved TTA-CU-based phototargeting of living cells [147]. As illustrated in Figure 14A, the micellar nanoparticles were composed of PdOEP and DPA incorporated into PLA−PEG polymeric micelles functionalized with DEACM-caged cRGDfK (c[R]GDfK—for details of each abbreviation see the Abbreviations section). Initially, PdOEP, DPA, and c[R]GDfK were all located in the PLA core of the micellar nanoparticles owing to their high hydrophobicity. Thus, the micellar nanoparticles exhibited a low binding ability to target cells (such as HUVECs and U87) in a PBS solutions since the binding groups were blocked. Upon irradiation with green light (530 nm), the short distances of PdOEP, DPA, and DEACM induced TTA-UC followed by FRET, and thus c[R]GDfK was found in high-efficiency photoinduced cleavage (Figure 14B). Consequently, cRGDfK was liberated from the core to the periphery of the nanoparticles. The cell-binding capability of these nanoparticles was greatly improved. This TTA-UC-assisted nanoparticle phototargeting provides a good reference for drug release and tumor treatment because the binding between functional groups and cell membranes could be finely regulated. Considering that green light was used in this study, further extending the operating wavelength to red light or even the NIR region represents an important topic for future research [148].

## 4. Conclusions and Perspective

Porphyrin-based TTA-UC systems integrate the merits of porphyrins and TTA-UC, and thus they possess unique advantages in many ways such as their long wavelength and low power of excitation light, strong absorption intensity, high penetrability, and tunable structures as well as tunable wavelengths. Based on the above characteristics, these porphyrin-based bimolecular upconversion systems were successfully used to trigger photochemical reactions for various applications, such as photoisomerizations for stimulus-responsive materials, photocatalytic organic synthesis, photoinitiated polymerization, photocatalytic degradation, and photochemical/photoelectrochemical water splitting. In the course of research, tremendous achievements have been witnessed by virtue of continuous molecular innovations and improved nanoarchitectonics. Photoreactions driven by visible and NIR light have been constantly developed. Apart from conventional bimolecular TTA-UC systems, a homomolecular TTA-UC process was also demonstrated by virtue of a ZnTPP molecule, and it was successfully utilized to initiate TTAP. Furthermore, the TTAP and 3D printing techniques were also strategically combined, and TTA-UC systems suitable for various printing modes were developed, representing a promising development in the 3D printing field of porphyrin-based TTA-TC systems. Aside from chemical synthesis, energy, and environment fields, porphyrin-based TTA-TC was also utilized to trigger photochemical reactions for biomedical applications, which plays an active role in phototargeting and drug release. In the processes of these TTA-UC-assisted photoreactions, oxygen quenching represents a persistent problem, and many strategies were previously applied to alleviate this problem, such as the use of supramolecular gel, nanocapsules, and an intercalation approach. The abovementioned results provide valuable references for the further design of TTA-UC-based photocatalysts and devices.

Although considerable progress has been made, the study on porphyrin-based TTA-UC for photoreactions is still in its infancy in comparison to well-investigated lanthanide-doped UCNPs. Therefore, further studies are required to expand its applications. First of all, further photoreactions are worth exploring. The pursuit of reactions with wider applicability, higher efficiency, higher selectivity, and faster dynamic regulation of stimulus-responsiveness provides motivation for synthetic chemists. Secondly, improving photocatalytic efficiency and stability synchronously is a very attractive topic considering that the existing methods can improve the molecular oxygen resistance but would reduce the upconversion efficiency. Moreover, simplifying the preparation of catalysts and devices is an important research direction. Finally, given the high-penetrating and low-scatting features of long-wavelength light as well as the good biocompatibility of porphyrin derivatives, explorations of further biochemical applications of porphyrin-based TTA-UC are deserving of the attention of scientists.

## Figures and Tables

**Figure 1 ijms-23-08041-f001:**
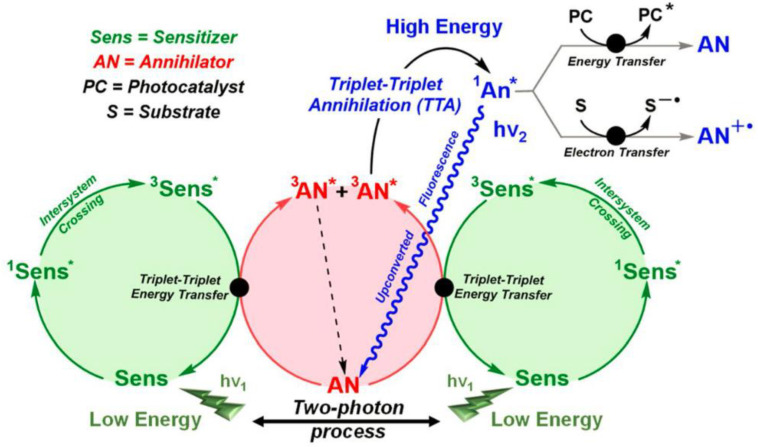
Schematic mechanisms of the TTA-UC process and its potential application for photochemical reactions, in which the symbol * refers to the excited state. Reprinted with permission from ref. [31]. Copyright 2022 Springer Nature.

**Figure 2 ijms-23-08041-f002:**
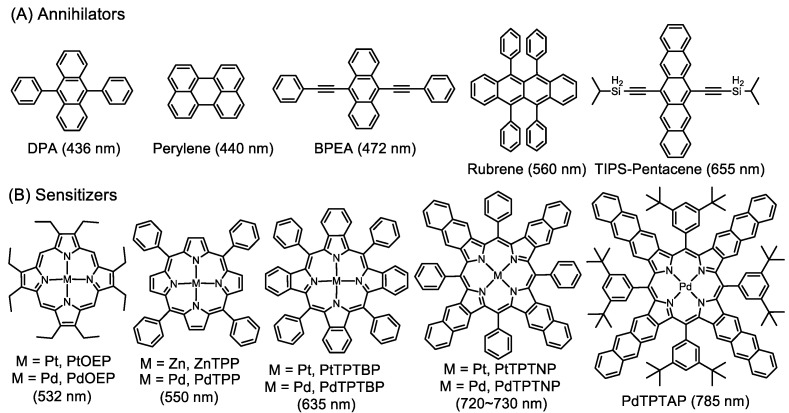
Chemical structures of commonly used annihilators and porphyrin-based sensitizers in TTA-UC and their corresponding emission and excitation wavelengths.

**Figure 3 ijms-23-08041-f003:**
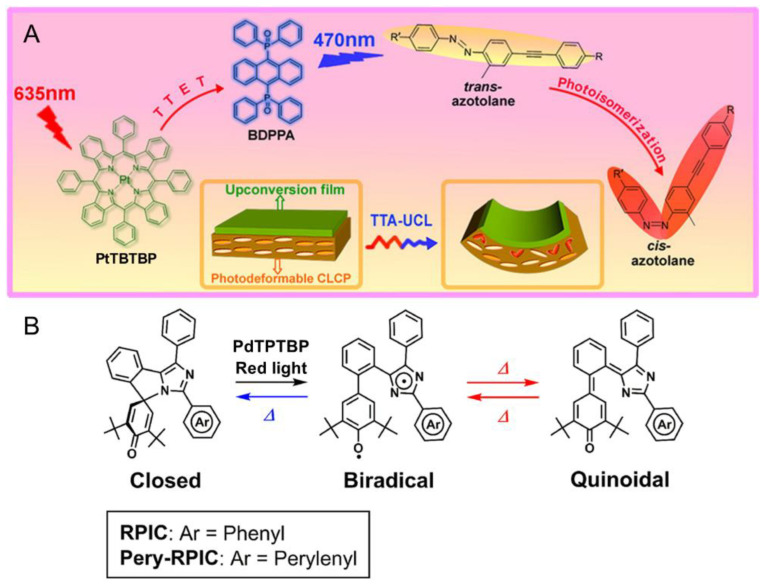
(**A**) Schematic illustration of liquid-crystal soft actuator modulated by red light based on TTA-UC. Reprinted with permission from ref. [76]. Copyright 2013 American Chemical Society. (**B**) Chemical structure of Pery-RPIC and its photoisomerization reactions driven by TTA-UC. Reprinted with permission from ref. [77]. Copyright 2019 American Chemical Society.

**Figure 4 ijms-23-08041-f004:**
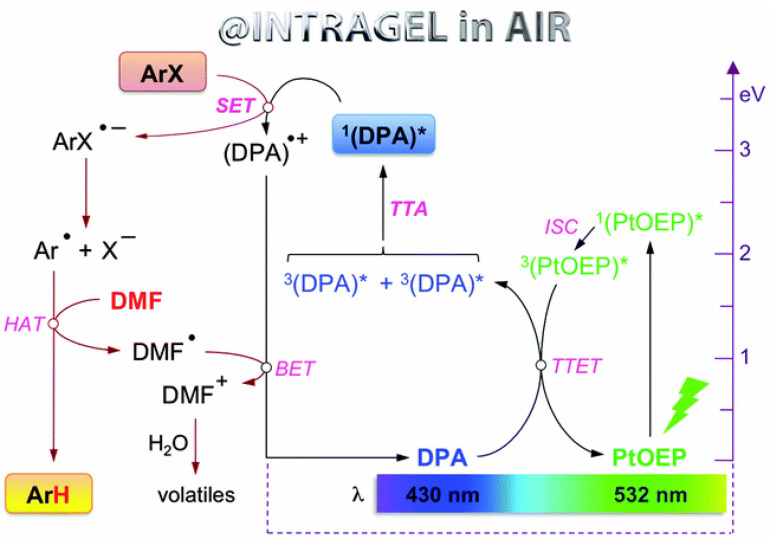
Proposed mechanism of the visible-light-triggered photoreduction of aryl halides under aerobic and room temperature conditions, in which the symbol * refers to the excited state. Reprinted with permission from ref. [86]. Copyright The Royal Society of Chemistry 2015.

**Figure 5 ijms-23-08041-f005:**
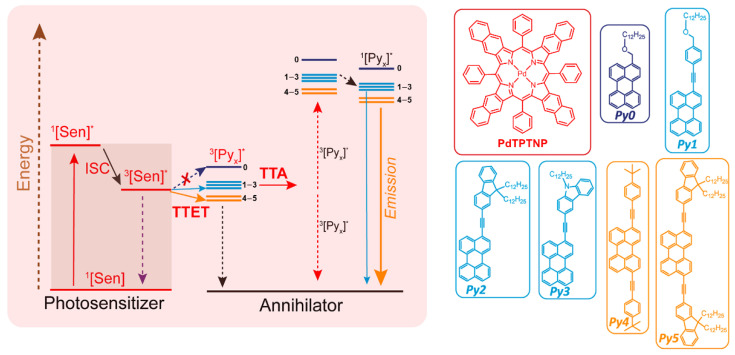
Schematic representation of the design for TTA-UC systems excited by NIR light via the optimization of annihilators (Py0−Py5), in which the symbol * refers to the excited state. Reprinted with permission from ref. [87]. Copyright 2020 American Chemical Society.

**Figure 6 ijms-23-08041-f006:**
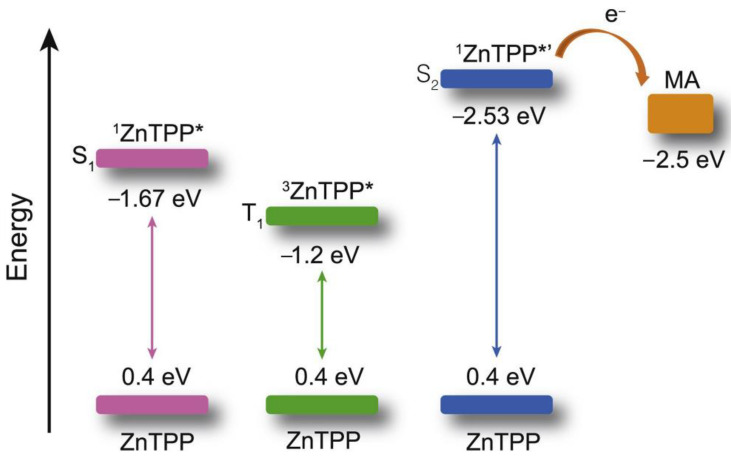
Schematic illustrations of the excited-state redox potentials of ZnTPP as well as monomer versus ferrocene^+^/ferrocene in CH_3_CN (the redox potential of TMPTA was hypothetically equal to that of MA) and the dynamic electron transfer between ^1^ZnTPP*’ and monomer, in which the symbol * refers to the excited state. Reprinted with permission from ref. [95]. Copyright 2020 Elsevier Ltd. All rights reserved.

**Figure 7 ijms-23-08041-f007:**
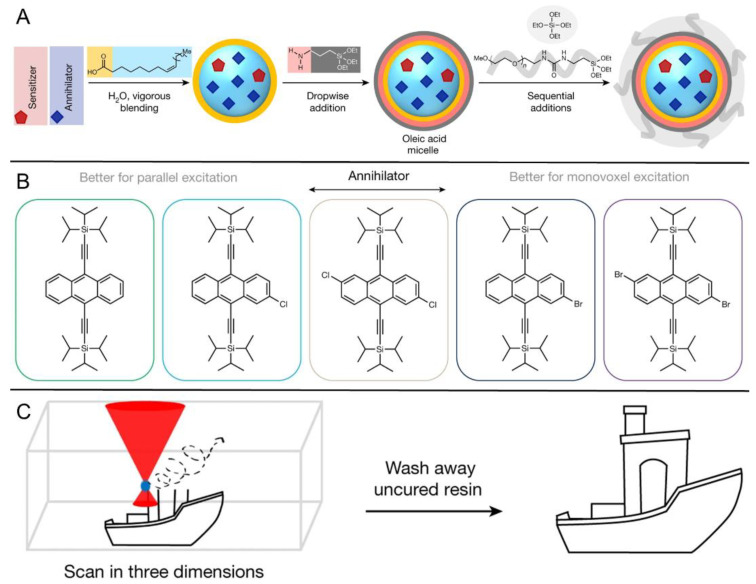
Schematic diagram of 3D printing based on TTAP: (**A**) Synthesis routes of the UCNP, (**B**) the chemical structures of selected annihilators, and (**C**) cartoon diagram of the 3D printing process in monovoxel printing mode. Reprinted with permission from ref. [70]. Copyright 2022 Springer Nature.

**Figure 8 ijms-23-08041-f008:**
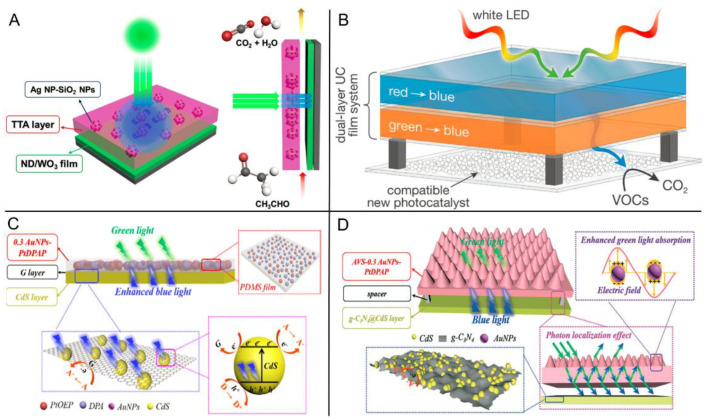
(**A**) Schematic representation of the plasmon-enhanced sub-bandgap photocatalysis via TTA-UC for acetaldehyde degradation. Reprinted with permission from ref. [113]. Copyright 2016 American Chemical Society. (**B**) Schematic diagram of ND/K-WO_3_ photocatalyst together with dual-layer TTA-UC systems for photocatalytic acetaldehyde degradation. Reprinted with permission from ref. [117]. Copyright 2019 American Chemical Society. (**C**) Schematic illustration of the photocatalytic mechanism of 0.3 AuNPs-PtDPAP/G/CdS. Reprinted with permission from ref. [118]. Copyright 2020 American Chemical Society. (**D**) Schematic diagram of the photocatalytic mechanism of the AVS-0.3 AuNPs-PtDPAP together with *g*-C_3_N_4_@CdS film. Reprinted with permission from ref. [119]. Copyright The Royal Society of Chemistry 2020.

**Figure 9 ijms-23-08041-f009:**
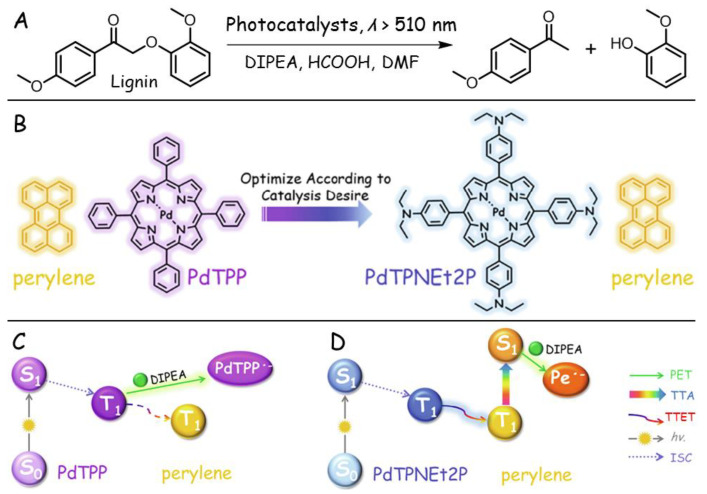
(**A**) Chemical reaction of the degradation of oxidized lignin model triggered by TTA-UC. (**B**) Molecular structures of the selected TTA-UC pairs: PdTPP/perylene pair and PdTPNEt2P/perylene pair. Schematic representations of the possible active species generation processes for (**C**) PdTPP/perylene pair and (**D**) PdTPNEt2P/perylene pair. Reprinted with permission from ref. [122]. Copyright 2022 Elsevier Ltd. All rights reserved.

**Figure 10 ijms-23-08041-f010:**
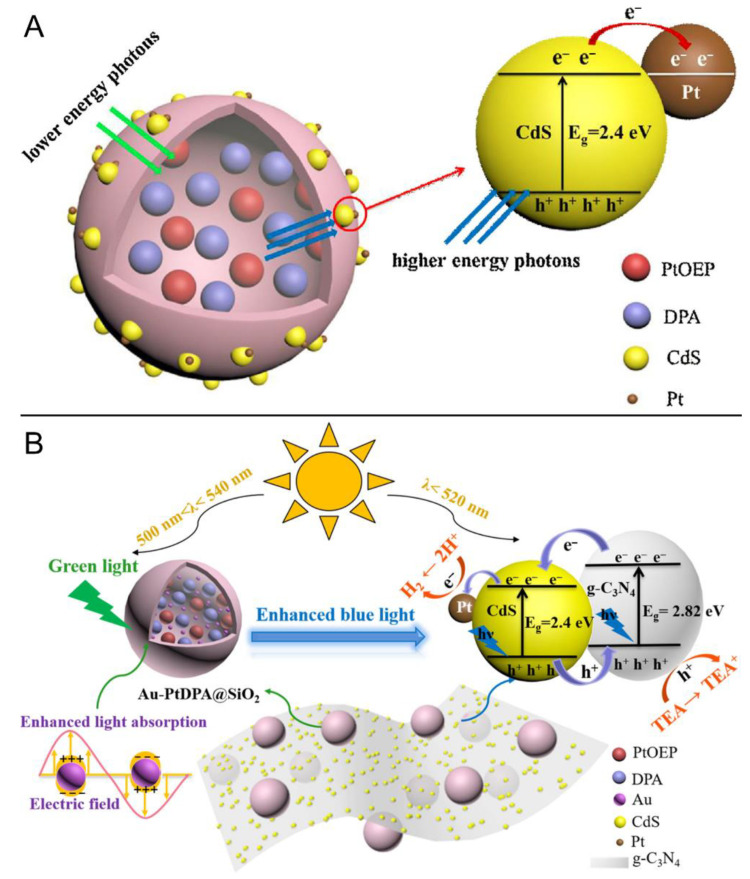
Schematic representations of the photocatalytic mechanisms of (**A**) PtDPAP@SiO_2_@CdS/Pt photocatalyst and (**B**) Au-PtDPAP@SiO_2_@*g*-C_3_N_4_-CdS photocatalyst based on TTA-UC process. Reprinted with permission from ref. [127]. Copyright 2017 Elsevier Ltd. All rights reserved. Reprinted with permission from ref. [128]. Copyright 2019 Elsevier Ltd. All rights reserved.

**Figure 11 ijms-23-08041-f011:**
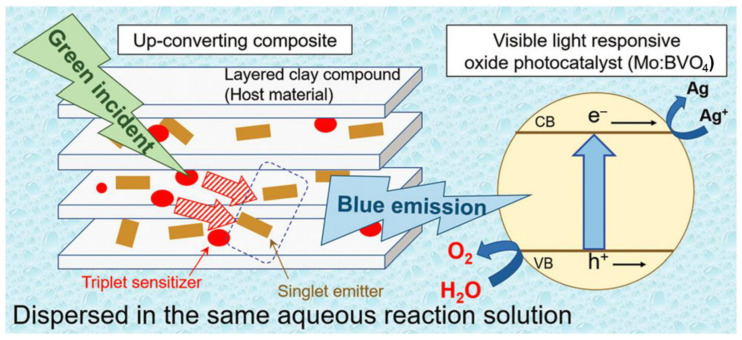
Schematic diagram of the photocatalytic OER triggered by TTA-UC from the composite photocatalyst containing PtOEP and DPA intercalated into a layered clay compound. Reprinted with permission from ref. [129]. Copyright The Royal Society of Chemistry 2021.

**Figure 12 ijms-23-08041-f012:**
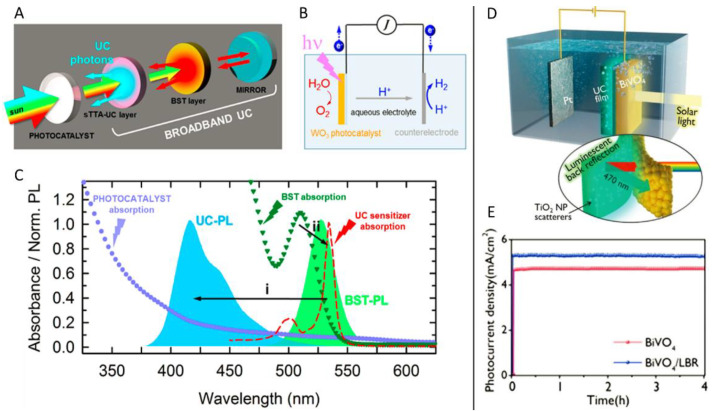
(**A**) Schematic diagrams of the multilayer broadband upconversion in (**B**) a classical photoelectrochemical water splitting cell with WO_3_, PtOET, DPA, poly(lauryl methacrylate) doped with CdSe/ZnS nanocrystals, and aluminum as photocatalyst, sensitizer, emitter, boosting layer (BST), and black mirror, respectively. (**C**) Absorption and fluorescence spectra of different layers. Reprinted with permission from ref. [135]. Copyright 2017 American Chemical Society. (**D**) Schematic diagram of the LBR-assisted photoelectrochemical water splitting of Mo:BiVO_4_ photocatalyst and (**E**) photocurrent-time responsive curves of Mo:BiVO_4_ photoanodes with and without the LBR layer. Reprinted with permission from ref. [136]. Copyright 2019 Wiley Online Library.

**Figure 13 ijms-23-08041-f013:**
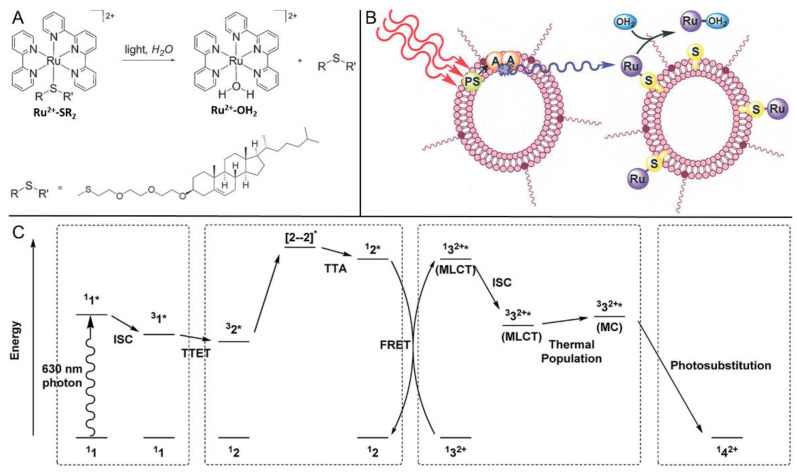
(**A**) Photochemical transformation of Ru^2+^-SRR’ into Ru^2+^-OH_2_ by blue light and (**B**) TTA-UC and photoinduced conversion processes in the lipid bilayer (PS and A refer to PdTPTBP photosensitizer and perylene acceptor). Reprinted with permission from ref. [145]. Copyright 2014 Wiley Online Library. (**C**) Schematic Jablonski diagram of the TTA-UC, FRET, and photosubstitution processes in the PEGylated liposomes upon 630 nm light excitation (the numbers 1, 2, 3 and 4 refer to PdTPTBP, perylene, Ru-SR_2_ and Ru-OH_2_ complexes, respectively). Reprinted with permission from ref. [146]. Copyright The Royal Society of Chemistry 2015.

**Figure 14 ijms-23-08041-f014:**
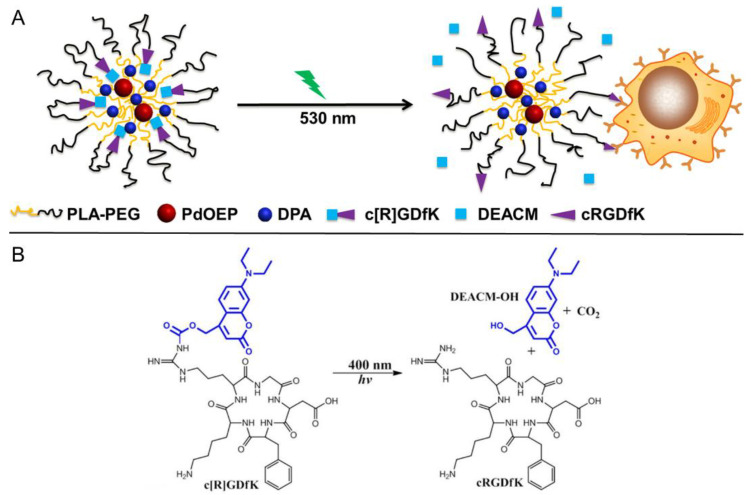
(**A**) Schematic illustration of the photocleavage of the polymeric micellar nanoparticles via TTA-UC and FRET. (**B**) Photocleavage reaction of c[R]GDfK (excitation at 400 nm). Reprinted with permission from ref. [147]. Copyright 2015 American Chemical Society.

**Table 1 ijms-23-08041-t001:** Summarized photoinduced organic synthesis reactions of porphyrin-based TTA-UC.

Entry	Sensitizer	Annihilator	Exemplary Reactions	Ref.
1	PtOEP	DPA	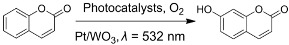	[84]
2	PdTPTBP	Perylene	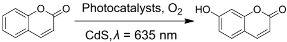	[85]
3	PtOEP	DPA	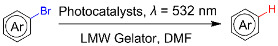	[86]
4	PtTPTNP	TTBP	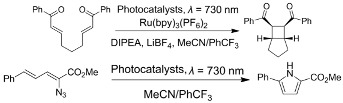	[69]
5	PdTPTNP	Py0~Py5	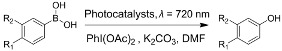	[87]

## Data Availability

Not applicable.

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
