# Peer review of "Recent Advances in the Photoreactions Triggered by Porphyrin-Based Triplet–Triplet Annihilation Upconversion Systems: Molecular Innovations and Nanoarchitectonicsâ€"

_ijms, 2022, doi:10.3390/ijms23148041_

Round 1
Reviewer 1 Report
It is a nice review paper. The topics described in this review article fit well with this journal. I basically recommend publication of this nice review in IJMS. For further improvements, I may suggest the following revisions.
1) The current title may give impression of usual progress review report. In order to increase impacts, some key features can be added to the title. These keywords would be Molecular Innovation and Nanoarchitectonics (as post-nanotechnology concept, see https://pubs.rsc.org/en/content/articlelanding/2021/nh/d0nh00680g). For example, the title like ... Recent Advances in the Photoreactions Triggered by Porphyrin-Based Triplet–Triplet Annihilation Upconversion Systems: Molecular Innovation and nanoarchitectonics … may sound more attractive.
2) Reference selection is basically OK but can be improved by addition of recent comprehensive papers on related topics (for example, https://www.journal.csj.jp/doi/10.1246/bcsj.20210114, https://pubs.acs.org/doi/10.1021/acsenergylett.1c01348)
Author Response
It is a nice review paper. The topics described in this review article fit well with this journal. I basically recommend publication of this nice review in IJMS. For further improvements, I may suggest the following revisions.
Reply: We greatly appreciate the reviewer’s positive comments on our review.
1) The current title may give impression of usual progress review report. In order to increase impacts, some key features can be added to the title. These keywords would be Molecular Innovation and Nanoarchitectonics (as post-nanotechnology concept, see https://pubs.rsc.org/en/content/articlelanding/2021/nh/d0nh00680g). For example, the title like ... Recent Advances in the Photoreactions Triggered by Porphyrin-Based Triplet–Triplet Annihilation Upconversion Systems: Molecular Innovation and nanoarchitectonics … may sound more attractive.
Reply: Thanks for the reviewer's excellent suggestion. The title of the article has been revised. Furthermore, the abstract, introduction as well as conclusion sections have been also adjusted to make the context more coherent. For example, a sentence “which would show us different mechanisms of TTA-UC and how to carry out reasonable molecular innovations and nanoarchitectonics to solve the problems existing in the practical application processes” was involved in the abstract section.
2) Reference selection is basically OK but can be improved by addition of recent comprehensive papers on related topics (for example, https://www.journal.csj.jp/doi/10.1246/bcsj.20210114, https://pubs.acs.org/doi/10.1021/acsenergylett.1c01348)
Reply: Thanks for the reviewer's excellent suggestion. More important progress and literatures on triplet–triplet annihilation upconversion are included in the introduction section, which is important to deepen the reader's understanding of this meaningful topic.
Reviewer 2 Report
In this review, Sun, Liu, and co-workers present a review of recent advances in the photoreactions triggered by porphyrin-based triplet-triplet annihilation upconversion systems. The manuscript is well written, and the topic and bibliography are timely and adequate. The length of the review is also convenient. My only remark is the quality of some Figures should be improved (for instance, some reprinted Figures are very detailed and contains information not discussed in the main text). After improving this readability aspect, I recommend this manuscript for publication in the International Journal of Molecular Sciences.
Author Response
In this review, Sun, Liu, and co-workers present a review of recent advances in the photoreactions triggered by porphyrin-based triplet-triplet annihilation upconversion systems. The manuscript is well written, and the topic and bibliography are timely and adequate. The length of the review is also convenient. My only remark is the quality of some Figures should be improved (for instance, some reprinted Figures are very detailed and contains information not discussed in the main text). After improving this readability aspect, I recommend this manuscript for publication in the International Journal of Molecular Sciences.
Reply: We greatly appreciate the reviewer’s positive comments on our review. Following the reviewer’s wonderful suggestion, several figures have been simplified and some non-essential images were removed, such as Figure 4 and Figure 7. For some schematics with complex structures, we selectively kept them unchanged as a sign of respect for the original authors and to provide great convenience for readers since these structures play irreplaceable roles in determining their functions.